# Interfering TAL effectors of *Xanthomonas oryzae* neutralize *R*-gene-mediated plant disease resistance

Zhiyuan Ji[1,2,*], Chonghui Ji[2,*], Bo Liu[2], Lifang Zou[1], Gongyou Chen[1] & Bing Yang[2]

Plant pathogenic bacteria of the genus *Xanthomonas* possess transcription activator-like effectors (TALEs) that activate transcription of disease susceptibility genes in the host, inducing a state of disease. Here we report that some isolates of the rice pathogen *Xanthomonas oryzae* use truncated versions of TALEs (which we term interfering TALEs, or iTALEs) to overcome disease resistance. In comparison with typical TALEs, iTALEs lack a transcription activation domain but retain nuclear localization motifs and are expressed from genes that were previously considered pseudogenes. We show that the rice gene *Xa1*, encoding a nucleotide-binding leucine-rich repeat protein, confers resistance against *X. oryzae* isolates by recognizing multiple TALEs. However, the iTALEs present in many isolates interfere with the otherwise broad-spectrum resistance conferred by *Xa1*. Our findings illustrate how bacterial effectors that trigger disease resistance in the host can evolve to interfere with the resistance process and, thus, promote disease.

---

[1] School of Agriculture and Biology/State Key Laboratory of Microbial Metabolism, Shanghai Jiao Tong University, Shanghai 200240, China. [2] Department of Genetics, Development and Cell Biology, Iowa State University, 1035C Roy J. Carver Co-Lab, Ames, Iowa 50011, USA. * These authors contributed equally to this work. Correspondence and requests for materials should be addressed to G.C. (email: gyouchen@sjtu.edu.cn) or to B.Y. (email: byang@iastate.edu).

Plant diseases are largely a consequence of molecular interactions between pathogens and their host plants and, when battles are won by the pathogens, they can inflict significant yield loss in crop production. Pathogenic microbes and their host plants have followed a 'zigzag' course that has co-evolved new virulence strategies in pathogens and counteracting resistance mechanisms in hosts[1]. Pathogenesis of many bacterial pathogens depends in part on the effector proteins translocated into host cells by a type-III secretion system[2]. Plants use diverse resistance (R) genes to recognize the cognate bacterial type-III effectors in a gene-for-gene manner, resulting in cultivar/race-specific disease resistance that prevents a state of disease susceptibility in plants[3]. Bacteria, in turn, diversify or inactivate the effector genes to evade the R gene recognition or evolve new effectors to suppress the resistance triggered by other distinct type-III effectors[4,5].

Transcription activator-like effectors (TALEs) represent the largest type-III effector family that are highly conserved at the nucleotide and amino acid levels[6], and are distinguishable by the varying number of central repeats of 34 amino acids and composition of the variable 12th and 13th amino acids of each repeat (so-called repeat variable di-residue). TALEs also contain characteristic nuclear localization motifs and transcription activation domain at their carboxyl termini[7] (Fig. 1a, exemplified by PthXo1). The repeat number and composition determine the specificity of each TAL effector for its DNA recognition in the promoter of host target gene, a feature that has spawned the development of TALE-based biotechnologies including TALE nucleases for genome editing[8].

TALEs play an important role in the pathogenesis of some Xanthomonas bacteria[7], including X. oryzae pv. oryzae (Xoo) and X. oryzae pv. oryzicola (Xoc), two pathogens that cause leaf blight by colonizing the vascular tissue and causing leaf streak by infecting the mesophyll tissue, respectively, in rice[9,10]. Bacterial TALEs target host genes of susceptibility (S gene) in a sequence-specific manner, resulting in enhanced bacterial growth and development of disease symptom[11]. To counteract such virulence strategy, host plants diversify the TALE-binding elements in the promoters of S genes, resulting in recessive R genes[12]. In addition, plants have also evolved so-called executor R genes to lure TALEs into triggering resistance similar to TALEs inducing host susceptibility[13]. Finally, in one case, tomato uses the nucleotide-binding leucine-rich repeat (NLR)-type R gene Bs4 to activate resistance in response to AvrBs4 independent of gene activation[14].

Here we demonstrate that Xa1, an NLR-type R gene in rice, initiates resistance including the hypersensitive response cell death (HR) in response to all tested full-length TALEs, whereas such resistance is suppressed by two groups of TALE variants expressed from previously annotated pseudogenes in Xoo and Xoc.

## Results

**Deletion of the Tal3 cluster triggers disease resistance.** PXO99A, a representative strain of Xoo, is virulent to a large number of rice varieties and contains 9 gene clusters totalling 19 individual TALE genes (Fig. 1b), some of which are important pathogenesis factors in bacterial blight of rice[15–18]. We generated a series of PXO99A mutant strains that are depleted of different and complete complements of TALE genes by sequentially deleting individual TALE gene clusters (Supplementary Fig. 1). Disease assays with those mutants on 36 rice varieties of different genetic backgrounds were performed to assess the pathogenesis role of each gene cluster. PXO99A is virulent to or compatible with 25 rice varieties, but avirulent to or incompatible with others

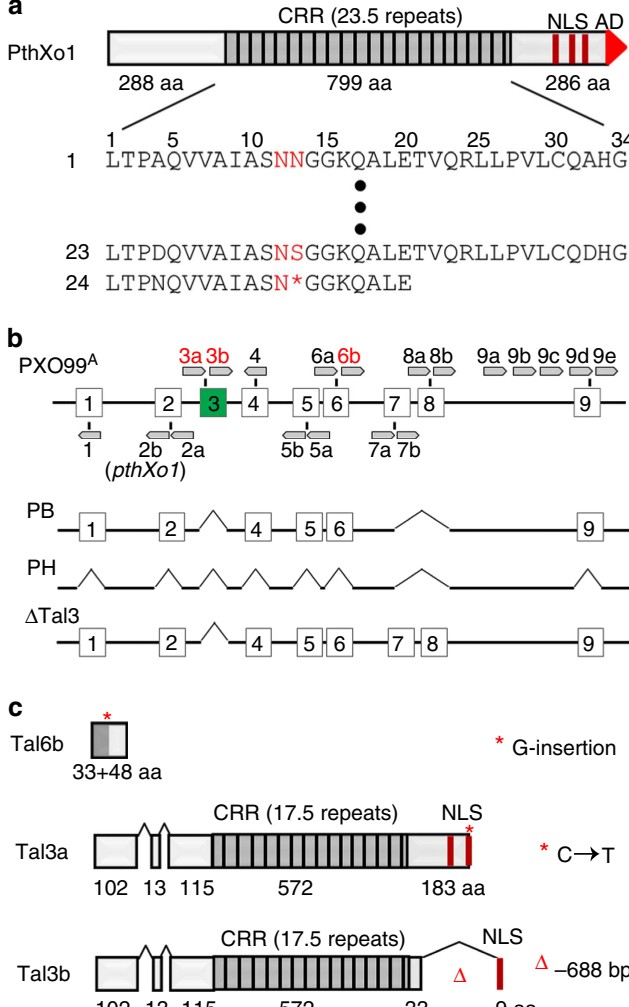

**Figure 1 | TALE and variants in PXO99A.** (a) Unique structure of TALEs typified by PthXo1. PthXo1 contains 23.5 nearly identical repeats of 34 amino acids in the central repeat region (CRR), with polymorphic repeat variable di-residue (RVDs; two residues, in red). Single letter is used for each amino acid and the asterisk (*) denotes the missing residue. The dots represent repeats not shown. The C terminus of PthXo1 contains three NLSs and the transcription activation domain (AD). (b) Nine clusters (enumerated boxes) of TALE genes (pentagons, not to scale) are located in the genome of PXO99A. Previously annotated pseudogenes (truncated genes or genes with premature stop codons, in red) are 3a, 3b and 6b. pthXo1 corresponds to 2b of cluster 2. PXO99A mutants PB, PH and ΔTal3 are cluster 3 + 7 + 8, all clusters and cluster 3 deleted (denoted by the carets, ˆ), respectively. (c) Three TALE variants deduced from the possible open reading frames (ORF) of three pseudogenes in PXO99A. The asterisk (*) denotes either the insertion (for a frame-shift) or the change of C to T for a stop codon, and the caret (ˆ) indicates the base pair deletion, relative to pthXo1.

due to some recessive (xa13) and dominant (Xa21, Xa27 and Xa23) R genes (Supplementary Table 1, column 1 and 2). In agreement with prior study[17], mutant of PXO99A with deletion of pthXo1-containing cluster lost the ability to cause disease in susceptible rice varieties. To our surprise, mutant PB with deletion of the cluster 3 (Tal3a/Tal3b) showed resistance in rice varieties IRBB1 and Kogyoku but not in other rice lines all susceptible to PXO99A (Fig. 1b and Supplementary Table 1).

PXO99A genome contains three TALE pseudogenes that have been annotated and previously reported[18]. Tal6b has a

1 bp insertion at the 97 bp position in the 5′-coding sequence. The nucleotide change may enable the gene to encode a new open-reading frame of 82 codons (Fig. 1c). *Tal3a* carries a premature stop codon due to a C to T change at the 3,013 bp position of the gene, probably encoding a protein with a C-terminal truncation of 103 amino acids, whereas *Tal3b* undergoes a large deletion (688 bp) at the 2,560 bp position relative to *Tal3a* and presumably encodes a product with 229 amino acids deleted and additional 10 amino acids due to a frame shift[18]. Both genes contain two deletions (129 and 45 bp) within the 5′-regions (Fig. 1c and Supplementary Figs 1 and 2). Tal3a and Tal3b, if expressed, are predicted to contain identical amino-termini, distinct central repetitive and C-terminal domains; both effectors contain the nuclear localization motifs but lack the transcriptional activation domains (Fig. 1c and Supplementary Fig. 2). Indeed, reverse transcription–PCR (RT–PCR) on bacterial RNA revealed the expression of both pseudogenes (Supplementary Fig. 3).

**iTALE Tal3a and Tal3b are virulence factors.** A mutant of PXO99[A] was constructed with only the cluster 3, containing the two TALE pseudogenes, deleted (ΔTal3; Fig. 1b) to assess the role of the pseudogenes in pathogenesis with other 17 TALE genes intact. ΔTal3 triggered HR in IRBB1 but not in IR24 when injected directly into the leaf blade (Fig. 2a). Similarly, ΔTal3 was able to cause disease in IR24 but not in IRBB1, on the basis of lesion length when the bacteria were introduced at the leaf tip (Fig. 2b). *Tal3a* and *Tal3b* were cloned and introduced, with an added FLAG epitope, individually to ΔTal3. Each clone enabled ΔTal3 to cause disease in IRBB1 comparable to the parent strain PXO99[A] (Fig. 2a,b). Western blotting probed with the anti-FLAG antibody showed the presence of Tal3a and Tal3b in the complementing strains of ΔTal3 (Supplementary Fig. 4). The results indicate that the TALEs *Tal3a* and *Tal3b* are not pseudogenes as

previously annotated but instead are expressed and function as TALE variants in PXO99[A] for virulence by interfering with the host resistance in IRBB1. Both effector variants and their relatives are referred to hereinafter as interfering TALEs (iTALEs).

**iTALEs interfere with *Xa1*-mediated resistance in rice.** IRBB1 and IR24 are near-isogenic rice lines for the *R* gene *Xa1* (ref. 19), which was identified as an NLR-type *R* gene from Kogyoku and IRBB1 with no cognate elicitor (or avirulence) gene identified yet[20]. To test whether the resistance to ΔTal3 and the suppressive effect of *Tal3a* and *Tal3b* were, in fact, specific to *Xa1* and not due to another gene in the IRBB1 background, the *Xa1* locus was PCR amplified from IRBB1 and transferred into the rice cultivar Kitaake, which is susceptible to PXO99[A] and ΔTal3. As expected, *Xa1* transgenic lines (n = 7), still susceptible to PXO99[A], were resistant to ΔTal3 in terms of HR and lesion length, but became susceptible to ΔTal3 in the presence of either *Tal3a* or *Tal3b* (Fig. 2c,d). The results demonstrate that PXO99[A] gains virulence by deploying its iTALE genes *Tal3a* and *Tal3b*, to mask the otherwise resistance in *Xa1*-containing plants.

**iTALEs need their unique structures to function.** Tal3a was characterized in more detail, to determine the requirement of each domain for activity of the iTALEs in *Xa1* context. Internal deletions of the central repeats resulted in three Tal3a variants that were expressed at a similar level in bacterial cells (Supplementary Fig. 5); all except one with 2.5 repeats retained the ability to suppress the resistance responses to ΔTal3 in IRBB1 and *Xa1* transgenic Kitaake (Fig. 3a,b). Similarly, Tal3a variants swapped with repeat domains from AvrXa7, AvrXa10 and PthXo1 were also able to suppress the resistance responses to ΔTal3 in IRBB1 (Supplementary Fig. 6). The results suggest the indispensability but not the repeat number and repeat variable

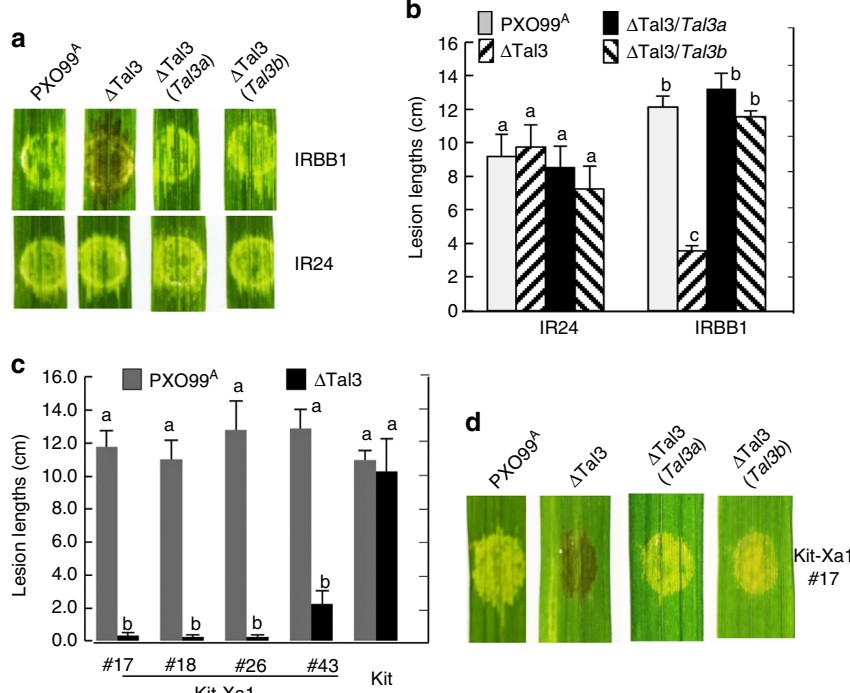

**Figure 2 | iTALE genes *Tal3a* and *Tal3b* interfere with *Xa1* resistance.** (**a**) Disease reactions of IRBB1 and IR24. Strains are indicated above the leaves. Brown colouration indicates HR and clearing spots indicate disease reaction. (**b**) Lesion lengths in IR24 and IRBB1 caused by Xoo strains as measured at 12 days post inoculation (DPI). (**c**) Lesion lengths in *Xa1* transgenic lines (n = 4) at 12 DPI. (**d**) *Xa1* transgenic line #17 showed HR (brown colouration) to ΔTal3 but disease reactions to ΔTal3 complementing strains and PXO99[A]. Photos were taken at 3 DPI. Error bars (s.d., n = 10, 3 replicates) with the same letter do not differ from each other at P < 0.05 (Tukey's test).

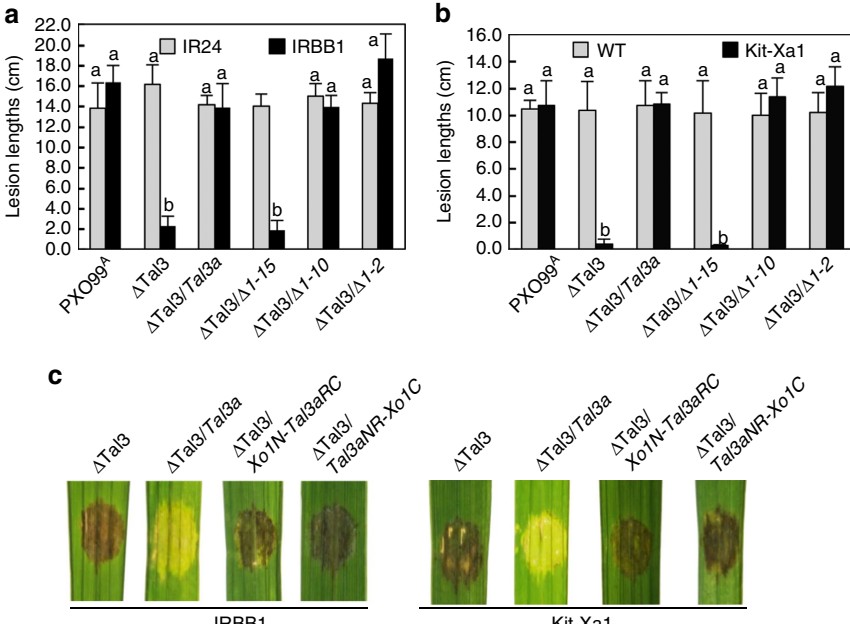

**Figure 3 | Unique domains inTal3a are required for its suppressive activity.** (**a**) Lesion lengths in IRBB1 and IR24 caused by Xoo strains as indicated below each column at 14 DPI. Δ1-15, Δ1-10 and Δ1-2 represent Tal3a with the N-terminal 15, 10 and 2 repeats deleted, respectively. (**b**) Lesion lengths in Kitaake (WT) and *Xa1* transgenic plants as assessed similarly in **a** but at 12 DPI. (**c**) Both Tal3a N-terminal and C-terminal structures are required for suppression of *Xa1*-mediated HR triggered by ΔTal3. Xo1N-Tal3aRC is a Tal3a variant containing PthXo1 N terminus and Tal3a central repetitive and C-terminal domains. Tal3aNR-Xo1C is a hybrid of Tal3a N-terminal and repetitive domains and PthXo1 C terminus. Photos were taken at 3 DPI. Error bars (s.d., *n* = 12, 2 replicates) with the same letter do not differ from each other at *P* < 0.05 (Tukey's test).

di-residue composition of the repeat domain for suppressive activity of Tal3a. The N terminus unique in two internal deletions in Tal3a was also tested for its contribution to the suppression. The N terminus of PthXo1 was swapped with the Tal3a corresponding region, the resultant Tal3a variant containing the N-terminal region of PthXo1 and Tal3a repetitive and C-terminal regions lost the ability to suppress the resistance triggered by ΔTal3 in IRBB1 and *Xa1* transgenic plants (Fig. 3c and Supplementary Fig. 7a). Similarly, swapping AvrXa7 N terminus into Tal3a also resulted in the loss of suppressive activity of Tal3a (Supplementary Fig. 7b). Likewise, Tal3a with the full-length C-terminal region of PthXo1 due to domain swapping lost its ability to suppress the resistance to ΔTal3 in IRBB1 and *Xa1* transgenic plants (Fig. 3c and Supplementary Fig. 7a). In their truncated C-termini, Tal3a still retains two nuclear localization signals (NLSs) and Tal3b acquires an NLS because of frame shift at its 3′-end (Supplementary Fig. 2); the NLS motifs were functional in directing the green fluorescent protein (GFP)-tagged Tal3a and Tal3b, to the nuclei of rice protoplasts. NLS mutations in Tal3a and Tal3b resulted in fluorescent signals present in cytosols, whereas the addition of the SV40 T-antigen NLS to the mutants restored the inclusion of fluorescence in the nuclei of rice protoplasts (Fig. 4a,b). When tested in plants, Tal3a and Tal3b variants with mutated NLS lost their abilities to suppress the resistance triggered by ΔTal3 in IRBB1; the addition of the SV40 T-antigen NLS restored their activities (Fig. 4c). The results indicate that the unique N- and C-terminal structures of Tal3a and Tal3b are essential and their NLSs (although not necessarily unique) are also needed for the iTALEs to interfere with the disease resistance controlled by *Xa1*.

***Xa1* activates resistance in response to full-length TALEs.** In the initial disease assay with the TALE cluster deletion mutants, the resistance in IRBB1 appeared when the clustered *Tal3a* and *Tal3b* were deleted and retained till remaining TALE clusters

were deleted. We surmised that *Xa1* might recognize TALEs and confer resistance against the pathogen only in the absence of iTALE genes. To test the hypothesis, we introduced TALE genes (*pthXo1*, *Tal4* and *Tal9d*) from PXO99[A] individually into PH, the TALE-free mutant of PXO99[A] (Fig. 1b). The resulting TALE-containing strains induced strong HR in *Xa1* transgenic Kitaake (Supplementary Fig. 8). Similarly, Tal3a and Tal3b variants that contain the full-length C termini due to domain swapping with PthXo1 also triggered HR in *Xa1* transgenic plants (Supplementary Figs 7a and 8) and IRBB1 (Supplementary Fig. 9). However, PthXo1 and AvrXa7 variants with the NLS mutated lost their abilities to trigger HR in IRBB1 and *Xa1* transgenic plants, whereas addition of SV40 T-antigen NLS restored their activities, suggesting a nuclear site of action of XA1/TALEs (Supplementary Fig. 10).

**Tal3a does not interfere with *Xa1* expression.** To determine whether *Xa1*, similar to *Xa27* and other executor *R* genes[15,16,21,22], recognizes TALEs through its promoter-specific transcription activation and iTALE overcomes resistance by suppressing *Xa1* induction, we made a construct expressing *Xa1* coding sequence under the promoter of a rice ubiquitin gene (*Os02g06640*). The Ubi:Xa1 transgenic Kitaake lines (*n* = 4) were completely resistant to ΔTal3 and the resistance was suppressed in the presence of *Tal3a* (Supplementary Fig. 11). The results indicate that the mode of action by iTALE is not through interference with *Xa1* transcription activation. To characterize the molecular role of iTALE in suppression of *Xa1* resistance in rice, three typical defense genes (peroxidase, *PBZ* and *PR1*) that are highly activated particularly during resistance response were assessed using the quantitative RT–PCR (qRT–PCR) approach. *Xa1* was induced slightly by wounding and bacterial infection, in agreement with the previous study[14]. In a contrast, in the incompatible interaction (*Xa1*/ΔTal3), all three defense genes were highly activated relative to non-infection and compatible

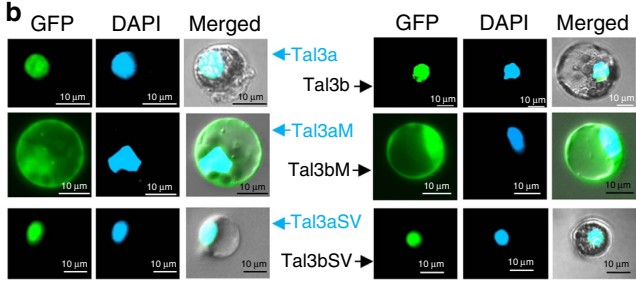

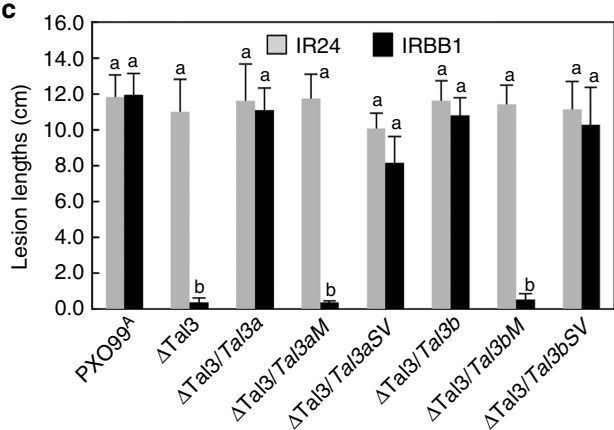

**Figure 4 | Nuclear localization motifs in Tal3a and Tal3b are required for their suppressive activities.** (**a**) Amino acids at the NLS and replacements are in red and underlined. (**b**) NLS motifs in Tal3a and Tal3b are functional to direct the GFP-tagged effectors into nuclei of rice protoplasts. (**c**) NLS motifs are required for Tal3a and Tal3b to interfere with the Tal3 triggered resistance in IRBB1. Lesion lengths of rice leaves caused by different strains as indicated below each column and measured at 12 DPI. Error bars indicate s.d. ($n = 9$, 3 replicates) with the same letters not different from each other at $P < 0.05$ (Tukey's test).

interaction (*Xa1*/ PXO99$^{\text{A}}$), whereas Tal3a suppressed the activations (Supplementary Fig. 12). The results indicate that iTALE overcomes *Xa1* resistance partially through suppressing the activation of defense genes.

**iTALE genes are prevalent among Xoo and Xoc isolates.** The indiscriminate recognition of TALEs by *Xa1* suggests that *Xa1* would be the broadest spectrum *R* gene known to date that is directed at bacterial blight and the only cloned rice-derived *R* gene to bacterial streak. To assess the resistance spectrum of *Xa1*, *Xa1* transgenic Kitaake plants were inoculated with 36 worldwide Xoo strains. The plants were resistant to only 7 field isolates but susceptible to the majority of 36 strains. The narrow resistance spectrum of *Xa1* is hard to reconcile to the notion that *Xa1* recognizes most, if not all, TALE genes and all examined Xoo strains contain large numbers (15 to 16) of TALE genes[18,23]. In fact, no *R* gene has ever been found and cloned for Xoc pathogen, of which strains contain the highest number (for example, 27 in BLS256) of TALE genes[24]. It is conceivable to attribute this to the

prevalence of iTALE genes in the majority of Xoo and Xoc populations. For example, *Tal3a* (referred to as type A) and *Tal3b* (type B) types of iTALE genes exist in three of four Xoo and all nine Xoc strains sequenced and well annotated to date (Supplementary Fig. 13)[18,23–25]. The known iTALE genes ($n = 18$) are highly conserved at the nucleotide level ($>99\%$ identity) and, if expressed, encode effectors that have nearly identical N termini in both types and nearly identical C termini in each type. The predicted iTALEs contain distinct central repeat domains (Supplementary Fig. 14).

We further assessed the prevalence of the two types of iTALE genes among 36 Xoo strains using a PCR approach with type-specific primers. Seven *Xa1*-incompatible strains contain either no detectable iTALE gene (three strains including AXO1947) or only type-B iTALE genes (four strains). AXO1947 has been sequenced and contains no iTALE gene[25], with which our PCR result from AXO1947 is in agreement. The remaining 29 *Xa1*-compatible strains indeed contain iTALE genes either of only type A (3 strains) or B (6 strains), or of both type A and B (19 strains; Supplementary Table 2). The four *Xa1*-incompatible strains, which include strain T7174 and contain only type-B iTALE genes, may either not express iTALE genes or express iTALE genes at a level not adequate to suppress *Xa1*-mediated resistance. To investigate this possibility, the iTALE gene *Tal3a* or *Tal3b* from PXO99$^{\text{A}}$ or the T7174 iTALE gene *Tal6* (type B) constructed under the *lacZ* gene promoter were introduced into T7174. Introduction of each plasmid-borne iTALE gene enabled T7174 to overcome the resistance in IRBB1 (Supplementary Fig. 15). The results appear to be in an agreement with our hypothesis. However, when transcripts of the type-B iTALE genes in bacterial strains that contain only type-B genes and are either *Xa1* incompatible (four strains) or compatible (six strains) were quantified using a qRT–PCR approach, no obvious correlation between expression levels of type-B iTALE genes in bacterial cells grown in medium and the disease phenotypes was observed (Supplementary Fig. 16). Therefore, the inability of the endogenous type-B iTALE genes to suppress the *Xa1* resistance by some Xoo strains needs further investigation in future.

We also cloned *Tal3* (type A) and *Tal6* (type B) from PXO86 of Xoo, *Tal11h* (type B) and *Tal12* (type A) from BXOR1, and *Tal5e* (type B) from RS105, two Xoc strains. All five iTALE genes, when transferred into ΔTal3, were functional in suppression of *Xa1*-mediated resistance (that is, in IRBB1 and *Xa1*-transgenic Kitaake) (Fig. 5a and Supplementary Fig. 17). Furthermore, for Xoc pathogen, when *Tal5e*, the only iTALE gene in RS105, was inactivated, the mutant was incompatible with IRBB1, and transfer of *Tal5e* or any of the four iTALE genes from Xoo enabled *Xa1* compatibility with the RS105 mutant (Fig. 5b,c and Supplementary Fig. 18). The results indicate that the type-A and type-B iTALE genes are evolutionarily conserved and functionally equivalent to contribute strain virulence by interfering with the *R* gene *Xa1*-mediated disease resistance against both Xoo and Xoc.

## Discussion

TALE-associated host *R* genes have been previously identified in rice (*Xa27*, *Xa10* and *Xa23*, *xa13*, *xa25* and *xa41*), tomato (*Bs4*) and pepper (*Bs3* and *Bs4C*); all of them, except one (*Bs4*), have been found to be involved in transcriptional activation (dominant *R* gene) or lack thereof (recessive alleles of the otherwise *S* genes) by the cognate full-length TALEs[14–16,22,26–30]. *Bs4*, a constitutively expressed *R* gene encoding an NLR protein in tomato, activates resistance including HR in response to the full-length TALE AvrBs4, as well as mutants derived from various truncations of C terminus and truncations of large portion of central repetitive and C-terminal regions that lack the nuclear

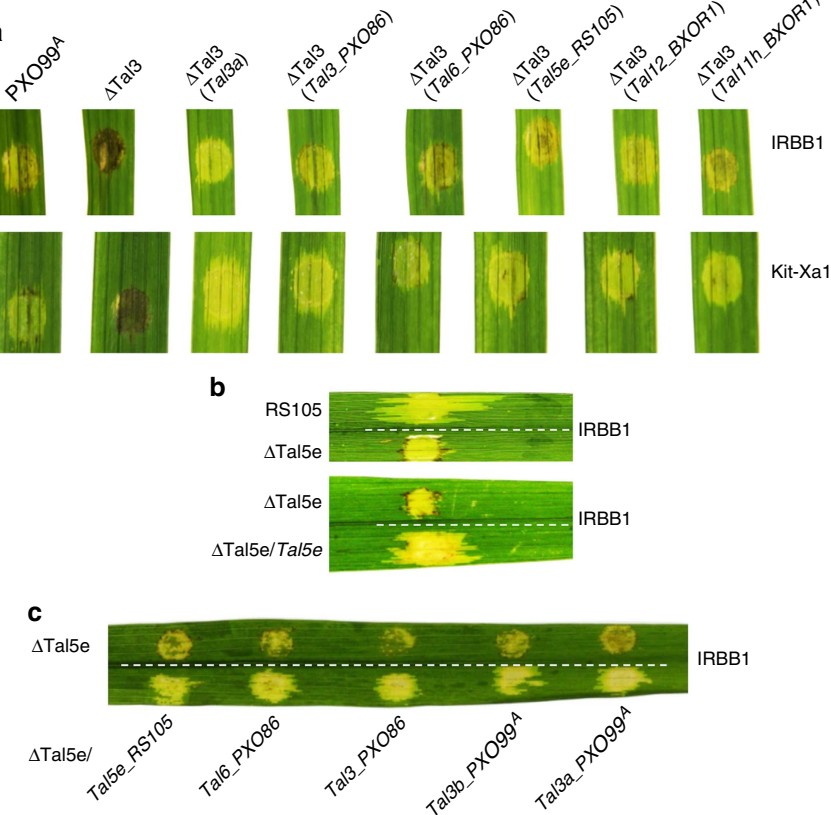

**Figure 5 | Suppressive activity of iTALE genes from Xoo and Xoc. (a)** iTALE genes from the Xoo PXO86, and Xoc RS105 and BXOR1 overcome *Xa1* resistance triggered by ΔTal3. **(b)** Inactivation of iTALE gene *Tal5e* enables the mutant (ΔTal5e) to trigger resistance in IRBB1. **(c)** iTALE genes from Xoo PXO86 and PXO99[A] restore the Xoc mutant (ΔTal5e) ability to cause disease in IRBB1. Photos were taken at 4 DPI.

localization and transcription activation domains of AvrBs4, suggesting cytoplasmic perception of AvrBs4 by Bs4 in tomato[14,31]. Most recently, a locus, *Xo1*, mapped in a 1.09 Mbp region in chromosome 4 of the heirloom rice variety Carolina Gold Selection, was found to activate resistance in response to various *X. oryzae* TALEs and TALE PthXo1 mutant with truncation of its C-terminal transcription activation domain[32]. Similar to *Xa1*, *Xo1*-mediated resistance is independent of the number of repeats (if >3.5 repeats) and the composition of the 12th and 13th amino acid residues of each repeat. It would be interesting to see whether *Xa1* and *Xo1* are the same gene or form a *Xa1 R* gene family. On the other hand, AvrBs4 can also be recognized by an executor *R* gene, *Bs4C* and trigger resistance in pepper. The recognition requires a match between the promoter element of *Bs4C* and the central repeats of AvrBs4 for tight expression of *Bs4C*, entailing a functionality of the full-length AvrBs4 (ref. 30). In contrast, *Xa1*, an NLR-type *R* gene unrelated to *Bs4*, recognizes all tested TALEs and initiates resistance in rice; the resistance elicitation requires the functional nuclear localization motif of TALEs. Furthermore, truncated TALEs (that is, iTALEs) as loss-of-function mutants avoid triggering *Xa1* resistance and are also as gain-of-function mutants able to suppress *Xa1* resistance triggered by full-length TALEs, in a way analogous to the dominant, negative regulators in host innate immunity.

Rice, evolutionarily speaking, has appeared to 'hit the jackpot' in the acquisition of an *R* gene that recognizes all or most TAL effectors. From the pathogen's stand point, exposure of multiple TALE targets to a cognate host *R* gene would be conundrum in that at least one TALE is critical for virulence in all strains.

We show that *X. oryzae* pathogens have evolved a potent adaptation to counteract the *Xa1*-controlled disease resistance in rice triggered by the large number of TALE genes of the pathogens using the very same genetic components. Understanding how one or two iTALEs efficiently mask the host immunity derived from recognition of multiple targets may enable engineering of more effective *R* genes that, for example, are less sensitive to the suppressive iTALE genes. *Xa1* and derived *R* genes may be an efficient genetic source to combat several other important crop diseases (for example, citrus canker and wheat blight) wherein the causative *Xanthomonas* agents possess TALE genes but not iTALE genes. In a broader light, our results suggest that the seemingly pseudogenes in a variety of bacterial genomes may warrant further examination.

## Methods

**Plant and bacterial materials.** Seeds of all rice varieties were kindly provided by the International Rice Research Institute, the U.S. National Small Grains Collection and collaborators. All plants were grown in growth chambers with photoperiod of 12 h, temperature of 28 °C daytime and 26 °C at night. *Escherichia coli* strains were grown in Luria–Bertani medium supplemented with appropriate antibiotics at 37 °C. All Xoo and Xoc strains were grown at 28 °C in nutrient broth with agar (NA) (1% peptone, 0.5% yeast extract, 1% sucrose, and 1.5% agar), nutrient broth without agar (NB), NA without sucrose, NA with 10% sucrose or TSA (10 g l$^{-1}$ tryptone, 10 g l$^{-1}$ sucrose and 1 g l$^{-1}$ glutamic acid). Antibiotics were used at the following concentrations, if required: cephalexin 10 μg ml$^{-1}$, kanamycin 25 μg ml$^{-1}$, ampicillin 100 μg ml$^{-1}$ and spectinomycin 100 μg ml$^{-1}$. Strains and plasmids used in this study are listed in Supplementary Table 3.

**TALE gene cluster deletion.** Suicide vector pKMS1 was used to generate PXO99[A] gene cluster deletion mutants using a method as described[33]. Nine clusters of TALE genes were sequentially deleted from PXO99[A] as indicated in Supplementary Fig. 1.

As DNA sequences of TALE genes are nearly identical, unique sequences flanking individual TALE clusters were chosen for knockouts. Based on the PXO99[A] genome sequence (NCBI accession, CP000967), two pairs of primers, ΦF1/ΦR1 and ΦF2/ΦR2 (Φ represents a TALE gene cluster), were used to amplify the upstream and downstream regions flanking the target TALE loci by using the PXO99[A] genomic DNA as the template (primer information is provided in Supplementary Table 4). The two PCR products for each cluster deletion were digested accordingly and cloned into the multiple cloning sites of pKMS1 and confirmed by sequencing for accuracy. The first round of mutagenesis was carried out on PXO99[A] targeting the duplicated clusters Tal7 and Tal8. Plasmid was transferred into the competent cells of PXO99[A] through electroporation and transformants were plated on the kanamycin NA lacking sucrose. Single colonies were transferred to NB medium without sucrose and incubated with shaking for 12 h at 28 °C. Bacterial cells were then plated on NA with 10% sucrose. Sucrose-tolerant colonies were duplicated on NA and kanamycin-containing NA plates. The kanamycin-sensitive colonies were screened by PCR (using primers Tal7/8F1-Tal7/8R2) and Southern blotting, to verify the deletion of the gene clusters 7 and 8 (see Supplementary Fig. 1). The mutant was used for the second round of mutagenesis targeting the cluster 3 of TALE genes. Similarly, sequential deletions were performed to complete the deletions of all nine TALE gene clusters (see Supplementary Fig. 1).

For Southern blotting, genomic DNA of PXO99[A] and its derived TALE mutants was extracted using the AxyPrep Bacterial Genomic DNA Miniprep Kit (Axygen, Hanzhou, China). DNA samples (3 μg) were digested with BamHI at 37 °C for 4 h, separated in 1.2% agarose gel through electrophoresis and transferred to Hybond N$^+$ nylon membranes (Millipore, Billerica, USA). The probe was made from a digoxigenin (DIG)-labelled 1368 bp SphI fragment containing the repetitive sequence of avrXa3 (GenBank accession number AY129298.1). Labelling, hybridization and detection procedures were performed by following the manufacturer's instruction (Roche, Sweden).

Tal5e deletion strain of RS105 of Xoc was similarly created. Specifically, primers Tal5RSF1-Tal5RSR1 and Tal5RSF2-Tal5RSR2 were used to generate the two homologous fragments for deletion of Tal5e.

### DNA manipulation and plasmid construction.
DNA manipulation and PCR were conducted according to standard protocols[34]. Plasmids were introduced by electroporation into X. oryzae and E. coli bacterial cells as described previously[35]. Primers for PCR were synthesized by Invitrogen Biotechnology Co., Ltd (Shanghai, China) and Integrated DNA Technologies (Coralville, IA, USA); PCR was performed with Ex-Taq (TakaRa Biotechnology, Dalian, China) and Phusion High-Fidelity DNA Polymerase (New England BioLabs, Ipswich, MA, USA).

Construction of genomic libraries for Tal3a, Tal3b and other iTALE genes was completed as following. Genomic DNA of PXO99[A] was digested with ClaI and separated in 1% agarose gel. DNA fragments of ∼4–6.5 kb were purified from the agarose gel and ligated into the ClaI-digested pBlueScript KS+ (Stratagene, La Jolla, CA, USA). The ligation reaction was transferred into E. coli DH5α cells. The library was screened for Tal3a and Tal3b using probe derived from the SphI fragment (repetitive region) of avrXa3. Candidate clones were sequenced for confirmation of Tal3a and Tal3b. To isolate the iTALE genes from PXO86 and RS105, genomic DNA was digested with BamHI and appropriate DNA fragments were purified from the agarose gel. The DNA fragments were subcloned into BamHI-digested pBlueScript KS+ and transferred into DH5α cells for screening of positive clones of iTALE genes Tal3 and Tal6 of PXO86 and Tal5e of RS105.

To construct the FLAG epitope-tagged Tal3a and Tal3b, primers Tal3aHFF–Tal3aHFR and Tal3bHFF–Tal3bHFR were used to amplify the 3'-regions of Tal3a and Tal3b, respectively. The purified PCR products were first digested using HincII and HindIII, and then along with BamHI–HincII fragments of Tal3a and Tal3b, ligated into the backbone of pZWavrXa7 (BamHI–HindIII digested), resulting in pZWTal3aF and pZWTal3bF, respectively. Both pZWTal3aF and pZWTal3bF were digested with HindIII and ligated into pHM1 (HindIII digested) to generate pHZWTal3aF and pHZWTal3bF. pHM1 is a plasmid replicable in Xanthomonas, whereas pZW version derived from pBlueScript is not replicable in Xanthomonas.

For construction of the internal central repeat deletions, pZWTal3aF was first completely digested with AatII, then partially with MscI; fragments in a range of 200 to 1,800 bp were recovered and ligated back to pZWTal3aF (digested with MscI-AatII). Clones with various sizes of repeat regions were selected and sequenced, to confirm the accuracy of deletions.

For domain swapping of avrXa7, avrXa10 and pthXo1 into Tal3a, the respective SphI central repetitive region of each gene was used to replace the corresponding region of Tal3a, resulting in pZWavrXa7a, pZWavrXa10a and pZWpthXo1a (see Supplementary Fig. 6a). The resulting plasmids were ligated into pHM1 at the HindIII cleavage site.

The full-length versions of Tal3a and Tal3b were constructed as following. The N-terminal and central repetitive domain coding regions were obtained with PstI and AatII from Tal3a and Tal3b, then swapped into the corresponding region of pZWpthXo1, resulting in pZWTal3aFL and pZWTal3bFL, respectively (see Supplementary Fig. 7a). The resultant plasmids were individually ligated into the HindIII-digested pHM1.

The chimeric Tal3a with the N terminus coding region of pthXo1 was constructed by cloning the BlpI–HindIII fragment from pZWTal3aF into the corresponding region of pZWpthXo1 (see Supplementary Fig. 7a). Similarly, BlpI–HindIII fragment of Tal3a was swapped into pZWavrXa7, resulting in gene encoding N terminus of AvrXa7 and the repetitive and C-terminal domains of Tal3a. Both pZW versions of Tal3a were subcloned into pHM1 by HindIII digestion and ligation.

To construct the NLS mutant of Tal3a, primers Tal3aMF1–Tal3aMR along with pZWTal3aF as template were used for the first round of PCR; the amplicon was used for the second round of PCR with primers Tal3aMF2–Tal3aMR. One mutation was incorporated into Tal3a in each round of PCR. The final PCR product was cloned back into pZWTal3aF by EcoRI and HindIII digestion, followed by ligation, resulting in pZWTal3aM. Primers Tal3aMF2–Tal3aMSVR along pZWTal3aM as template were used to add the SV40 NLS coding sequence into Tal3aM through a PCR approach and subsequently cloning through EcoRI/HindIII digestion and ligation. Similarly, Tal3bM (NLS mutant) was constructed. Primers Tal3HincIIF–Tal3bMR along with pZWTal3bF as template were used to incorporate NLS mutant sequence into Tal3b via a PCR approach. The PCR amplicon was digested with HincII and HindIII, and ligated back into pZWTal3bF, resulting in pZWTal3bM. The addition of SV40 NLS coding sequence was carried out using PCR with primers Tal3HincIIF–Tal3bSVR plus Tal3bM as template, followed by HincII and HindIII digestion and DNA ligation, resulting in pZWTal3bSV. The resulting plasmids were sequenced for the accuracy of PCR-amplified regions. All pZW versions of plasmids were ligated into pHM1 through HindIII digestion and ligation.

GFP-tagged Tal3a and Tal3b were constructed using PCR and standard cloning approaches. Primers GFPKp-F and GFPBam-R along with an enhanced GFP (eGFP) template were used to PCR amplify the GFP coding region. The PCR product cloned into pGEM-T vector through A/T cloning and sequenced for accuracy. The eGFP coding region was cut out with KpnI and BamHI. The digested eGFP DNA fragment along with BamHI–HindIII fragments of Tal3aF, Tal3aM, Tal3aSV, Tal3bF, Tal3bM and Tal3bSV was ligated under the CaMV 35S promoter and Nos terminator in pUC19 (digested by KpnI and HindIII), respectively.

Gene encoding the PthXo1 NLS mutation was constructed by swapping the whole 3'region (813 bp) downstream of AatII recognition site in pZWpthXo1 with a gBlock synthesized from the Integrated DNA Technologies. The gBlock encoding the three NLS mutations was used to replace the corresponding region of pthXo1 at AatII and HindIII cleavage sites using Gibson cloning method. Similarly, gBlock encoding the NLS mutations and additional SV40 NLS was swapped into the corresponding region of pthXo1 in pZWpthXo1. The NLS mutation and addition of the SV40 NLS for avrXa7 in pZWavrXa7M123 (referred to as avrXa7M) and pZWavrXa7SV40 (referred to as avrXa7SV), respectively, were described[17]. The pZW versions of pthXo1 each were subcloned into pHM1 at the HindIII restriction sites.

To clone iTALE genes Tal11h and Tal12 from BXOR1, primers BXOR1F and BXOR1R that are complementary to the flanking regions of both genes were used to amplify the respective fragments from the genomic DNA. The amplicons were cloned into pHM1 (BamHI digested) directly through Gibson cloning. The accuracy of cloning was confirmed via DNA sequencing.

### Transient gene expression and microscopy.
The mesophyll protoplasts of rice cultivar Kitaake were isolated and transfected as described[36]. Rice protoplasts transfected with eGFP-Tal3a, eGFP-Tal3b and their NLS mutants were observed 36 h post transfection using a Leica SP5 X MP confocal/multiphoton microscope at the ISU Confocal and Multiphoton Facility. Fluorescence images were acquired at 522–572 nm (eGFP) and 358–461 nm (4,6-diamidino-2-phenylindole).

### Genotyping of Xoo strains for the presence of iTALE genes.
Primers Tal3aF1–Tal3aR1 and Tal3bF1–Tal3bR1 were used along with the genomic DNA of individual strains for detection of the Tal3a- and Tal3b-type iTALE genes, respectively.

### RT–PCR analysis.
Bacterial RNA was extracted from Xoo cells grown in XOM2 medium at 28 °C and using TRI Reagent Solution (ThermoFisher Scientific, Waltham, MA, USA) as described[37]. One microgram of total RNA was treated with DNase I (ThermoFisher Scientific), to eliminate the DNA contamination and used for complementary DNA synthesis by using iScript cDNA Synthesis kit (Bio-Rad, Hercules, CA, USA) with random 9-mers following the user's manual. cDNA derived from 50 ng of RNA was used for each reaction of semi-quantitative PCR (qPCR). Semi-qPCR for Tal3a and Tal3b gene expression in PXO99[A] was performed by using gene-specific primers Tal3aF2–Tal3aR2 and Tal3bF1–Tal3bR1, respectively. Ribosomal 16S RNA expression was used as an internal control with gene-specific primers (16SrRNA-F and 16SrRNA-R). Real-time qRT–PCR was performed on Strategene's Mx4000 multiplex using iQ SYBR Green Supermix (Bio-Rad). Relative quantification was based on the expression levels of Tal3b versus the 16S rRNA gene by using the $2^{Cq(ref)-Cq(target)}$ method, a variation of the Livak method[38], to determine the expression ratio.

For plant transcript detection, RNA was extracted from leaves inoculated with bacteria as specified in the text. One microgram of total RNA was first treated with DNase I (ThermoFisher Scientific) and used for cDNA synthesis by using the

iScript cDNA Synthesis kit (Bio-Rad). cDNA derived from 25 ng of total RNA was used for each real-time PCR, which was performed on Strategene's Mx4000 multiplex qPCR system using iQ SYBR Green Supermix (Bio-Rad). The gene-specific primer sequences are provided in Supplementary Table 4. The average threshold cycle (Ct) was used to determine the fold change of gene expression. As an internal control, rice actin gene was used. The $2^{\Delta\Delta Ct}$ method was used for relative quantification[37].

**Rice transformation.** For construction of *Xa1*, primers Xa1F1–Xa1R1 and Xa1F2–Xa1R2 were used to amplify two fragments from *Xa1* locus in IRBB1. The two overlapping amplicons were joined and inserted into SacI site in pCAMBIA1300 using Gibson Assembly Master Mix (New England BioLabs). *Xa1* with ubiquitin promoter was constructed with a synthetic DNA fragment (751 bp) of the 5′-end and a PCR amplicon (4,664 bp) of the 3′-end of *Xa1* under the *rice ubiquitin 2* gene promoter in pCAMBIA1300. Both cDNA clone pUbi:Xa1 and genomic clone p1300-Xa1 were electroporated into *Agrobacterium tumefaciens* strain EHA105. Calli from immature embryos of Kitaake were initiated and transformed by using *A. tumefaciens* as described[39]. Transgenic plants were genotype with primers (Xa1F3 and U) located at the 3′ of *Xa1* and in the backbone of pCAMBIA1300, respectively.

**Disease assays.** Hypersensitive cell death response (HR) and virulence assays were conducted as described previously[40]. Briefly, Xoo strains were grown in NB with appropriate antibiotics at 28 °C. Bacterial cells were collected from culture by low-speed (4,000 r.p.m.) centrifugation, washed twice and suspended in sterile water. The suspensions were adjusted to an optical density of 0.5 at 600 nm and were used to infiltrate into leaves of rice seedlings (about 3 weeks old) with the needleless syringe to assess the strain ability to trigger HR in plants. The cells of the same concentration were also used to inoculate two fully expanded leaves of five to ten adult plants (about 2 months old) using the leaf-tip clipping method, to evaluate the strain ability to cause disease or trigger resistance in plants by measuring the lesion lengths. Similarly, inoculum of Xoc was infiltrated into rice leaves using the needleless syringe to measure the rice reactions (susceptible or resistant). The disease assays were performed at least twice. One-way analysis of variance statistical analyses were performed on all measurements. The Tukey's honest significant difference test was used for post analysis of variance pair-wise tests for significance, set at 5% ($P < 0.05$).

**Data availability.** The data that support the findings of this study are available within the paper and its Supplementary Information files.

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

## Acknowledgements

This research was supported by the National Science Foundation of China research grant 31230059 (G.C.), the State Key Basic Research and Development Project of China grant

2012CB114003 (G.C.), the Special Fund for Agro-scientific Research in the Public Interest of China 201303015 (G.C.), the China Scholarship Council (Z.J., as a joint PhD student) and the US National Science Foundation research grants 2012-1238189 and IOS-1258103 (B.Y.). We thank Frank White (University of Florida) for critical reading of the manuscript.

## Author contributions

Z.J., C.J., B.L. and L.Z. performed the experiments. Z.J., G.C. and B.Y. conceived the experiments. Z.J., C.J. and B.Y. prepared the manuscript with input from all other co-authors.

## Additional information

**Competing financial interests:** The authors declare no competing financial interests.

**Publisher's note**: 
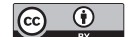

