## [Peer Review File · Nature Communications]

Reviewers' comments:

Reviewer #1 (Remarks to the Author):

This reviewer congratulates the authors to this outstanding piece of work! Systematic mutageneses of all tal gene clusters in the *Xanthomonas oryzae* model strain PXO99A allowed uncovering the presence and activity of tal gene variants, called iTALEs, which had hitherto commonly considered to be non-functional pseudogenes. The key discoveries of the manuscript are:

(i) the rice resistance gene Xa1 mediates recognition of TAL effectors independent of their DNA-binding specificity, and

(ii) most strains of *X. oryzae* evolved iTALEs which share unique structural characteristics (e.g. distinct N-terminal domain) and suppress Xa1-mediated resistance.

This is unexpected data and has tremendous implications for future resistance breeding against xanthomonads relying on tal gene activity, i.e. not only for rice pathogens but also for pathogens of barley, cassava, citrus and cotton, among others. The experiments are carefully designed and data have been well interpreted. Even if the mechanism of TAL effector recognition and defense suppression is not clear yet (if so, the paper would have made it as a full article into Nature, this reviewer supposes), the data allow generation of testable hypotheses and will stimulate future work. Probably because of the format of Nature Communications the discussion is rather short and models to understand the molecular mechanism of recognition and suppression are not presented. Interestingly, a related paper was just published in The Plant Journal (accepted manuscript online: 20 May 2016) by Lindsay Triplett and colleagues, and this reviewer wonders whether Xa1 and Xo1 (Triplett et al.) are the same gene? Even with restricted space, maybe the authors could briefly comment on this. Otherwise, the manuscript is complete and a pleasure to read.

Minor comments:

Page 7, second to last line: lacZ gene with small initial, not LacZ.

Page 8, second line: BXOR1, not BROX1.

Page 8, third line: Tal5e (type B).

Supp. Figure 13: Tal12_BXOR1, not Tal12h_BXOR1.

Reviewer #2 (Remarks to the Author):

TAL effectors (TALEs) promote virulence by upregulating susceptibility genes. Resistance can evolve by bringing defense genes under this upregulation (eg Bs3), or by evolving alleles of S genes that are refractory to this upregulation (eg xa13, a recessive resistance). Resistance can also evolve via NLR detection of TAL effectors (Xa1, Bs4).

The authors report interesting findings suggesting that truncated TALEs interfere with NLR-dependent defense responsiveness to TALEs, and they term these iTALEs.

Specifically the authors show that iTALEs in Xoo and Xoc suppress the function of the otherwise broad-spectrum rice Xa1 NLR gene. They support their claims as follows.

Xoo strain PX099 is virulent on Xa1-containing rice, but deletion of TAL cluster 3 encoding Tal3a and Tal3b results in loss of virulence. Tal3a and b contain NLS but lack TA domain; either restores virulence to the deletion strain. Suppression activity requires the TALE repeat units, the correct N terminus and an NLS. The mechanism of interference is not by inhibiting Xa1 expression. iTALEs seem to be widespread in Xoo strains that overcome Xa1 and are absent in strains that do not.

Broadly the data support this interesting and novel contribution to the field.

Can the authors please comment on the following 2 points?

"We further assessed the prevalence of the two types of iTALE genes among thirty-six *X. o. pv. oryzae* strains using a PCR approach with type-specific primers." Surely absence of PCR evidence is not evidence of absence. Are there no sequenced strains that are resisted by Xa1? Could they sequence at least one Xa1-resisted strain for a less biased approach to investigating the presence of iTALEs?

How widespread are iTALE like genes in eg Solanaceae, Brassicaceae, cassava, banana etc - infecting strains?

Reviewer #3 (Remarks to the Author):

This manuscript describes an interesting and novel virulence strategy evolved in two important and related *Xanthomonas* bacterial pathogens of rice. In this strategy, truncated versions of *Xanthomonas* TAL effectors (called iTALE) interfere with recognition of full length TALE by the previously cloned plant R gene Xa1, a gene that recognizes and activates resistance (an HR response) to a large range of TALE. The manuscript represents a large amount of work that establishes the importance of iTALE in interfering with resistance governed by Xa1. The paper does not identify or propose mechanism, and the text throughout needs to be carefully edited to reflect that the mechanism is still not known. The text also needs some editing for clarity and English (see below for a few examples). Finally, there are inconsistencies that the authors need to more adequately address prior to publication. In particular, they propose a hypothesis for why some bacterial strains have iTALE, but yet do not exhibit the suppression of Xa1 mediated resistance, but they do not really address this hypothesis in the experiments they report (see point 18 below).

Specifics:

1. Abstract and elsewhere: "neglected" is not needed as an adjective for "pseudogenes" because it really is meaningless....neglected by who? Researchers? The plant? The pathogen?
2. Abstract: "require unique N-termini, truncated C-termini, and nuclear localization motifs for their suppressive activities"...reword...the paper shows a requirement for nuclear localization motifs...but not UNIQUE nuclear localization motifs as the sentence structure implies.
3. In the abstract and throughout the paper, the acronym NBS-LRR is used without ever being defined; the acronym NLR is now more commonly used to refer to this kind of protein
4. Throughout the paper, the authors refer to two pathogens (referring to *X. oryzae pv. oryzae* and *X. oryzae pv. oryzicola*). For those who don't understand this nomenclature, somewhere in the introduction they should add a sentence or two as to the biology that makes these two bacterial pathogens different.
5. "In addition, plants have also evolved so-called "executor" R genes to lure TAL effectors into triggering resistance in a way that the pathogens direct expression of S genes." This sentence is not clear...what is the connection the authors are trying to make?
6. The next sentence is also not clear, i.e., what 'transcriptional functionality' is referred to for AvrBs4
7. A recently published paper that reports broad recognition of diverse TALE by a different rice resistance gene (Xo1) should be cited, as in that paper, they report recognition similar to pepper Bs4 in rice (Triplett et al. 2016. Plant Journal). This paper needs to be discussed here and in other places to put Xa1 and Xo1 into context for the readers.
8. For general readers not familiar with the nomenclature of TALE and what 'clusters' contain which TALE genes, it is very hard to follow the text on the deletion mutants created in strain PXO99, especially when the labeling changes in some of the figures. It would be helpful if Supplemental Fig 1a

were integrated into the main body of the text, as it is essential to understanding the mutants used in the study. Mutant and genetic complementation constructs should also be included in this figure.

9. Due to the novelty of the iTALEs, a more descriptive gene structure image of Tal3a, Tal3b, and Tal6b that indicates the positions of NLS, AD, CRR, #Repeats, and how they compare to the reference pthXo1 TALE would be useful.

10. The same detail (See suggestion 9) should be included in Supplemental Fig 5a.

11. A diagram for the Xo1N-Tal3aRC and Tal3aNR-Xo1C (all chimeras) would be helpful to provide detail as to where PthXo1/Tal3a begins and where it ends - this for all constructs.

12. The list of plants tested in Supplemental Table 1 is impressive, and represents a lot of work, but what is important about the different varieties tested? Do they have different reactions to the strain PXO99?

13. Text: deletion of cluster 3 "started" to show resistance....clearer just to say "showed resistance"?

14. Tal6b mentioned as a third iTALE, but no further description/study. Is it predicted to give a truncated product? Does it have an effect? It was deleted later, so was its possible effect masked by the deletion in cluster 3? Is it transcribed?

15. "and the resistance were reversed by introduction of either Tal3a or Tal3b to Δ Tal3 (Fig. 1d, e)." Besides the need to fix the English in this section, the authors' use of the term 'reversed' is interesting because it implies the resistance is activated, then reversed....is the resistance blocked, masked, overwhelmed, or activated and then reversed? The mechanism of iTALEs will be very interesting! But for now, if you don't know mechanism, be careful how you describe it throughout the paper.

16. "The results indicate that the unique N- and C-terminal structures of Tal3a and Tal3b are essential for the iTALEs to interfere with the disease resistance controlled by Xa1." Add in that nuclear localization signals (although not necessarily unique) are also needed.

17. "The four Xa1-incompatible strains, including strain T7174, that contain B type iTALE genes may either not be expressed or are expressed at a level not adequate to efficiently suppress Xa1-mediated resistance." This last section is incomplete, and confusing. If I understand correctly, there are four strains that give the HR on Xa1 plants, but these strains have iTALEs, so are inconsistent with the authors' hypothesis. One of the strains (T7174) is the original strain used to identify Xa1 function for cloning (i.e., that strain activates HR on Xa1 rice).

18. These authors propose that the strains do not suppress Xa1 function because of weak or no expression of their iTALEs (a reasonable hypothesis). So, they clone and put the iTALE from those strains under a lacZ promoter and introduce into different strains, including T7174, and show that the iTALE suppress the HR. This suggests the iTALE can function in those strains, and is consistent with expression being important, but it still does not explain why the native iTALEs in the strains are not functioning (and does not test their hypothesis). To show this, they would need to measure expression of the iTALE in the native strains or under a native promoter?

19. The authors switch between showing data for Tal3a and Tal3b throughout the paper...why?

20. Figure 3. Missing data. The effect of Tal3aM and Tal3aSV in nuclei of rice protoplast images should be included in this figure.

Discussion:

21. Include discussion of Xa1 with the recently published paper on TALE and a different resistance gene (Xo1) should be cited, as they showed a recognition similar to Bs4 in rice (Triplett et al. 2016)

22. Saying "pseudogenes" in quotes may be misinterpreted. Make it clear that these genes are not pseudogenes, and are only thought to be, since they code a functional protein

Methods:

23. The Methods section is very hard to follow, particularly when trying to figure out how different clones were constructed. Some examples:

a. According to the accession number pointed to, the genome sequence referred to is to a strain called PXO99A....was this the strain used, or was PXO99 used for the bulk of the work in the paper?

- b. "cluster deletion were restricted accordingly"....what is the meaning of "restricted" here?
 - c. what is avrXa3, and why was it used here?
 - d. Typo: "As an internal control, rice action gene was used."
 - e. "genomic region of Xa2 was PCR-amplified with primers (Xa2F1/R1) and genomic DNA of IRBB2....". What is Xa2, why was it cloned, and of what relevance is it to this paper? The methods is the only place this gene was mentioned.
- Disease assays section: misspelling: descried should be described.
- f. Missing statistics in all figures; were they done? (error bars do not compare treatments)
 - g. Disease assay section does not have the sample size (number of plants used per treatment) listed.

Reviewer #1:

This reviewer congratulates the authors to this outstanding piece of work! Systematic mutageneses of all tal gene clusters in the *Xanthomonas oryzae* model strain PXO99A allowed uncovering the presence and activity of tal gene variants, called iTALEs, which had hitherto commonly considered to be non-functional pseudogenes. The key discoveries of the manuscript are:

- (i) the rice resistance gene *Xa1* mediates recognition of TAL effectors independent of their DNA-binding specificity, and
- (ii) most strains of *X. oryzae* evolved iTALEs which share unique structural characteristics (e.g. distinct N-terminal domain) and suppress *Xa1*-mediated resistance.

This is unexpected data and has tremendous implications for future resistance breeding against xanthomonads relying on tal gene activity, i.e. not only for rice pathogens but also for pathogens of barley, cassava, citrus and cotton, among others. The experiments are carefully designed and data have been well interpreted. Even if the mechanism of TAL effector recognition and defense suppression is not clear yet (if so, the paper would have made it as a full article into Nature, this reviewer supposes), the data allow generation of testable hypotheses and will stimulate future work. Probably because of the format of Nature Communications the discussion is rather short and models to understand the molecular mechanism of recognition and suppression are not presented. Interestingly, a related paper was just published in The Plant Journal (accepted manuscript online: 20 May 2016) by Lindsay Triplett and colleagues, and this reviewer wonders whether *Xa1* and *Xo1* (Triplett et al.) are the same gene? Even with restricted space, maybe the authors could briefly comment on this. Otherwise, the manuscript is complete and a pleasure to read.

Our response:

We appreciate the positive comments from this reviewer!

It is not known whether *Xo1* described by Triplett et al. and *Xa1* are the same gene as *Xo1* was only mapped in a 1.09 Mbp region where *Xa1* is also located. We briefly discussed the *Xo1* work in the revised manuscript in the Discussion: “Most recently, a locus, *Xo1*, mapped in a 1.09 Mbp region in chromosome 4 of the heirloom rice variety Carolina Gold Selection, was found to activate resistance in response to various *Xanthomonas oryzae* TALEs and TALE PthXo1 mutant with truncation of its C-terminal transcription activation domain³¹. Like *Xa1*, *Xo1*-mediated resistance is independent of the number of repeats (if > 3.5 repeats) and the composition of the 12th and 13th amino acid residues of each repeat. It would be interested to see whether *Xa1* and *Xo1* are the same gene or form a *Xa1* R gene family. ”

Minor comments:

Page 7, second to last line: lacZ gene with small initial, not LacZ.

Our response:

We changed LacZ to *lacZ*

Page 8, second line: BXOR1, not BROX1.

Our response:

Changed to BXOR1

Page 8, third line: Tal5e (type B).

Our response:

Added type B: Tal5e (type B)

Supp. Figure 13: Tal12_BXOR1, not Tal12h_BXOR1.

Our response:

Changed to Tal12_BXOR1

Reviewer #2

TAL effectors (TALEs) promote virulence by upregulating susceptibility genes. Resistance can evolve by bringing defense genes under this upregulation (eg Bs3), or by evolving alleles of S genes that are refractory to this upregulation (eg xa13, a recessive resistance). Resistance can also evolve via NLR detection of TAL effectors (Xa1, Bs4).

The authors report interesting findings suggesting that truncated TALEs interfere with NLR-dependent defense responsiveness to TALEs, and they term these iTALEs.

Specifically the authors show that iTALEs in Xoo and Xoc suppress the function of the otherwise broad-spectrum rice Xa1 NLR gene. They support their claims as follows.

Xoo strain PX099 is virulent on Xa1-containing rice, but deletion of TAL cluster 3 encoding Tal3a and Tal3b results in loss of virulence. Tal3a and b contain NLS but lack TA domain; either restores virulence to the deletion strain. Suppression activity requires the TALE repeat units, the correct N terminus and an NLS. The mechanism of interference is not by inhibiting Xa1 expression. iTALEs seem to be widespread in Xoo strains that overcome Xa1 and are absent in strains that do not.

Broadly the data support this interesting and novel contribution to the field.

Can the authors please comment on the following 2 points?

We further assessed the prevalence of the two types of iTALE genes among thirty-six *X. o. pv. oryzae* strains using a PCR approach with type-specific primers." Surely absence of PCR evidence is not evidence of absence. Are there no sequenced strains that are resisted by Xa1? Could they sequence at least one Xa1-resisted strain for a less biased approach to investigating the presence of iTALEs?

Our response:

One strain, AXO1947, one of the 36 strains tested and shown to be resistant by *Xa1*, has been recently sequenced. The sequenced genome contains no iTALE gene, with which our results is in an agreement. The reference (#30) has been cited with a brief discussion added in the revised manuscript)

How widespread are iTALE like genes in eg Solanaceae, Brassicaceae, cassava, banana etc - infecting strains?

Our response:

We still could not find any Tal3a or Tal3b-like iTALE genes in other *Xanthomonas* strains that infect Solanaceae, Brassicaceae, cassava, and banana as host plants, based on the database sequences.

Reviewer #3

This manuscript describes an interesting and novel virulence strategy evolved in two important and related *Xanthomonas* bacterial pathogens of rice. In this strategy, truncated versions of *Xanthomonas* TAL effectors (called iTALE) interfere with recognition of full length TALE by the previously cloned plant R gene *Xa1*, a gene that recognizes and activates resistance (an HR response) to a large range of TALE. The manuscript represents a large amount of work that establishes the importance of iTALE in interfering with resistance governed by *Xa1*. The paper does not identify or propose mechanism, and the text throughout needs to be carefully edited to reflect that the mechanism is still not known. The text also needs some editing for clarity and English (see below for a few examples). Finally, there are inconsistencies that the authors need to more adequately address prior to publication. In particular, they propose a hypothesis for why some bacterial strains have iTALE, but yet do not exhibit the suppression of *Xa1* mediated resistance, but they do not really address this hypothesis in the experiments they report (see point 18 below).

Specifics:

1. Abstract and elsewhere: "neglected" is not needed as an adjective for "pseudogenes" because it really is meaningless....neglected by who? Researchers? The plant? The pathogen?

Our response:

“neglected” was deleted.

2. Abstract: "require unique N-termini, truncated C-termini, and nuclear localization motifs for their suppressive activities"...reword...the paper shows a requirement for nuclear localization motifs...but not UNIQUE nuclear localization motifs as the sentence structure implies.

Our response:

We reworded the sentence to: "iTALs require unique N- and truncated C-termini, and also nuclear localization motifs for their suppressive activities."

3. In the abstract and throughout the paper, the acronym NBS-LRR is used without ever being defined; the acronym NLR is now more commonly used to refer to this kind of protein.

Our response:

We defined and changed NBS-LRR into NLR.

4. Throughout the paper, the authors refer to two pathogens (referring to *X. oryzae* pv. *oryzae* and *X. oryzae* pv. *oryzicola*). For those who don't understand this nomenclature, somewhere in the introduction they should add a sentence or two as to the biology that makes these two bacterial pathogens different.

Our response:

We added one sentence to reflect the different biology of the two diseases: "...two pathogens that cause leaf blight by colonizing the vascular tissue and causing leaf streak by infecting the mesophyll tissue, respectively, in rice."

5. "In addition, plants have also evolved so-called "executor" R genes to lure TAL effectors into triggering resistance in a way that the pathogens direct expression of S genes." This sentence is not clear...what is the connection the authors are trying to make?

Our response:

We changed the sentence to: "In addition, plants have also evolved so-called "executor" R genes to lure TALEs into triggering resistance similar to TALEs inducing host susceptibility."

6. The next sentence is also not clear, i.e., what 'transcriptional functionality' is referred to for AvrBs4.

Our response:

We changed the sentence to "Finally, in one case, tomato uses the NLR type *R* gene *Bs4* to activate resistance in response to AvrBs4 independent of gene activation."

7. A recently published paper that reports broad recognition of diverse TALE by a different rice resistance gene (*Xo1*) should be cited, as in that paper, they report recognition similar to pepper

Bs4 in rice (Triplett et al. 2016. *Plant Journal*). This paper needs to be discussed here and in other places to put *Xa1* and *Xo1* into context for the readers.

Our response:

We added in the revised manuscript: “Most recently, a locus, *Xo1*, mapped in a 1.09 Mbp region in chromosome 4 of the heirloom rice variety Carolina Gold Selection, was found to activate resistance in response to various *Xanthomonas oryzae* TALEs and TALE PthXo1 mutant with truncation of its C-terminal transcription activation domain³¹. Like *Xa1*, *Xo1*-mediated resistance is independent of the number of repeats (if > 3.5 repeats) and the composition of the 12th and 13th amino acid residues of each repeat. It would be interested to see whether *Xa1* and *Xo1* are the same gene or form a *Xa1* R gene family.”

8. For general readers not familiar with the nomenclature of TALE and what 'clusters' contain which TALE genes, it is very hard to follow the text on the deletion mutants created in strain PXO99, especially when the labeling changes in some of the figures. It would be helpful if Supplemental Fig 1a were integrated into the main body of the text, as it is essential to understanding the mutants used in the study. Mutant and genetic complementation constructs should also be included in this figure.

Our response:

We added a new figure (**Figure 1**) and changed Supplementary Figure 1 to show the unique features of TALEs as exemplified by PthXo1 (Fig. 1a), cluster composition of TALE genes in PXO99^A and deletions (Fig. 1b), and gene structures of three pseudogenes as previously annotated in PXO99^A (Fig. 1c; Sup. Fig. 1a). However, we added the mutant and genetic complementation constructs separately in other figures associated with the corresponding disease phenotypes (e.g., Sup. Fig. 6; Sup. Fig. 7).

9. Due to the novelty of the iTALEs, a more descriptive gene structure image of Tal3a, Tal3b, and Tal6b that indicates the positions of NLS, AD, CRR, #Repeats, and how they compare to the reference pthXo1 TALE would be useful.

Our response:

We added a new text figure (Fig. 1c) (see above).

10. The same detail (See suggestion 9) should be included in Supplemental Fig 5a.

Our response:

We changed the Supp. Fig. 5a.

11. A diagram for the Xo1N-Tal3aRC and Tal3aNR-Xo1C (all chimeras) would be helpful to provide detail as to where PthXo1/Tal3a begins and where it ends - this for all constructs.

Our response:

We added a new supplementary figure (Supp. Fig. 7a) to reflect that.

12. The list of plants tested in Supplemental Table 1 is impressive, and represents a lot of work, but what is important about the different varieties tested? Do they have different reactions to the strain PXO99?

Our response:

We initially wanted to test how different rice varieties that are either resistant or susceptible to PXO99^A react to the TALE cluster deletion mutant, so we included as many varieties as possible. Indeed to our surprise, *Xal*-containing rice reacted differently to the cluster 3 deletion mutant, the basis for this study.

Yes, those rice varieties have different reactions to PXO99^A. They are either resistant or susceptible to the pathogen as shown by different lesion lengths in column 2 in the Supplementary Table 1.

13. Text: deletion of cluster 3 "started" to show resistance....clearer just to say "showed resistance"?

Our response:

We changed the wording as suggested.

14. Tal6b mentioned as a third iTALE, but no further description/study. Is it predicted to give a truncated product? Does it have an effect? It was deleted later, so was its possible effect masked by the deletion in cluster 3? Is it transcribed?

Our response:

No, Tal6b was not mentioned or defined as an iTALE in our study, instead only Tal3a and Tal3b were due to their function. The 1-bp insertion in Tal6b may result in a new ORF of 81 amino acids. We added this information in Figure 1c and in the revised manuscript. However, we did not pursue it further and, therefore, could not answer some of the reviewer's questions.

15. "and the resistance were reversed by introduction of either Tal3a or Tal3b to Δ Tal3 (Fig. 1d, e)." Besides the need to fix the English in this section, the authors' use of the term 'reversed' is interesting because it implies the resistance is activated, then reversed....is the resistance blocked, masked, overwhelmed, or activated and then reversed? The mechanism of iTALEs will be very interesting! But for now, if you don't know mechanism, be careful how you describe it throughout the paper.

Our response:

We revised the sentence to “As expected, *Xa1* transgenic lines (n=7), still susceptible to PXO99^A, were resistant to Δ Tal3 in terms of HR and lesion length, but became susceptible to Δ Tal3 in presence of either *Tal3a* or *Tal3b* (Fig. 2c, d).”

16. "The results indicate that the unique N- and C-terminal structures of Tal3a and Tal3b are essential for the iTALEs to interfere with the disease resistance controlled by *Xa1*." Add in that nuclear localization signals (although not necessarily unique) are also needed.

Our response:

We revised the sentence as suggested to “The results indicate that the unique N- and C-terminal structures of Tal3a and Tal3b are essential and their nuclear localization signals (although not necessarily unique) are also needed for the iTALEs to interfere with the disease resistance controlled by *Xa1*.”

17. "The four *Xa1*-incompatible strains, including strain T7174, that contain B type iTALE genes may either not be expressed or are expressed at a level not adequate to efficiently suppress *Xa1*-mediated resistance." This last section is incomplete, and confusing. If I understand correctly, there are four strains that give the HR on *Xa1* plants, but these strains have iTALEs, so are inconsistent with the authors' hypothesis. One of the strains (T7174) is the original strain used to identify *Xa1* function for cloning (i.e., that strain activates HR on *Xa1* rice).

Our response:

We revised the sentence to “The four *Xa1*-incompatible strains, which include strain T7174 and contain only B type iTALE genes, may either not express iTALE genes or are express iTALE genes at a level not adequate to efficiently suppress *Xa1*-mediated resistance.”

18. These authors propose that the strains do not suppress *Xa1* function because of weak or no expression of their iTALEs (a reasonable hypothesis). So, they clone and put the iTALE from those strains under a lacZ promoter and introduce into different strains, including T7174, and show that the iTALE suppress the HR. This suggests the iTALE can function in those strains, and is consistent with expression being important, but it still does not explain why the native iTALEs in the strains are not functioning (and does not test their hypothesis). To show this, they would need to measure expression of the iTALE in the native strains or under a native promoter?

Our response:

We investigated the RNA levels of the type B iTALEs in some strains that contain only type B iTALE genes and are either compatible or incompatible to *Xa1* as the reviewer suggested by using a RT-PCR approach. However, we did not find obvious correlation between level of Tal3b gene expression in bacterial cells grown in medium and the disease phenotype. We provide the results in Supplementary Figure 16 and added one sentence to reflect the implication of the

results (“Therefore, the inability of the endogenous *Tal3b* to suppress the *Xa1* resistance by some Xoo strains needs further investigation in future.”).

19. The authors switch between showing data for Tal3a and Tal3b throughout the paper...why?

Our response:

We presented data for both Tal3a and Tal3b as they are two different types of iTALEs and function similarly but may slightly differently to suppress the *Xa1*-mediated resistance. For example, all strains that contain only Tal3a are virulent to *Xa1*, while not all strains that containing only Tal3b are virulent to *Xa1*.

20. Figure 3. Missing data. The effect of Tal3aM and Tal3aSV in nuclei of rice protoplast images should be included in this figure.

Our response:

We included the nuclear localization data of all three versions of Tal3a and Tal3b in the revised manuscript (new Figure 4b).

Discussion:

21. Include discussion of *Xa1* with the recently published paper on TALE and a different resistance gene (*Xo1*) should be cited, as they showed a recognition similar to *Bs4* in rice (Triplett et al. 2016).

Our response:

We added the new reference of which we were not aware during our manuscript preparation. We also discussed the work in our revised manuscript.

22. Saying "pseudogenes" in quotes may be misinterpreted. Make it clear that these genes are not pseudogenes, and are only thought to be, since they code a functional protein.

Our response:

We deleted the quotes as suggested.

Methods:

23. The Methods section is very hard to follow, particularly when trying to figure out how different clones were constructed. Some examples:

a. According to the accession number pointed to, the genome sequence referred to is to a strain called PXO99A....was this the strain used, or was PXO99 used for the bulk of the work in the paper?

Our response:

We changed PXO99 to PXO99^A in the revised manuscript as it is the reference strain sequenced.

b. "cluster deletion were restricted accordingly"....what is the meaning of "restricted" here?

Our response:

We changed “restricted” to “digested” throughout the manuscript.

c. what is *avrXa3*, and why was it used here?

Our response:

avrXa3 a member of TALE gene family (GenBank accession no. AY129298.1) and was used to generate a probe for Southern blotting.

d. Typo: "As an internal control, rice action gene was used."

Our response:

We corrected “action” to “actin”.

e. "genomic region of *Xa2* was PCR-amplified with primers (*Xa2F1/R1*) and genomic DNA of IRBB2....". What is *Xa2*, why was it cloned, and of what relevance is it to this paper? The methods is the only place this gene was mentioned.

Disease assays section: misspelling: *descried* should be *described*.

Our response:

We Deleted the *Xa2* related text.

We also corrected the misspelling of “described”.

f. Missing statistics in all figures; were they done? (error bars do not compare treatments)

Our response:

Statistics were added in all figures with measurements.

g. Disease assay section does not have the sample size (number of plants used per treatment) listed.

Our response:

We used 5 to 10 plants with two leaves for each treatment. The information is added in the Method.

REVIEWERS' COMMENTS:

Reviewer #2 (Remarks to the Author):

I am broadly satisfied with the author's responses to my comments and to those of the other reviewers. Their improved text will need some proof reading e.g.

It would be interested to see whether Xa1 and Xo1 are the same gene or form a Xa1 R gene family.

should be "It would be interesting"

there are several such examples of problems with the English in their responses

Reviewer #3 (Remarks to the Author):

The manuscript is very much improved, and apart from some needed English editing in a few places, should be good to go. Congratulations to the authors on an excellent study!

A few comments:

1. Is it appropriate to refer to these iTALEs as pseudogenes? By definition, pseudogenes are functionless. In fact, this paper's major finding is that these genes are not pseudogenes.
2. Fig 1a – PthXo1 is said to have 24 repeats above the diagram; this should be changed to 23.5 repeats, which is shown with the diagram and the repeat sequence below the diagram.
3. Should NLR be defined in the introduction, not just in the abstract?
4. It is unclear which cluster pthXo1 is from in Fig 1b. Can you indicate?
5. Comment: Fig 4b is impressive – much improved from the previous version.
6. Thanks for including Supplementary Fig. 16: some clarification: is this figure showing expression of the plasmid-borne Tal3b from PXO99A? and the native "B-type" iTALEs from each of the other strains?

Responses to the reviewers:

Reviewer #2 (Remarks to the Author):

I am broadly satisfied with the author's responses to my comments and to those of the other reviewers. Their improved text will need some proof reading e.g.

It would be interested to see whether Xa1 and Xo1 are the same gene or form a Xa1 R gene family.

should be "It would be interesting"

there are several such examples of problems with the English in their responses.

Our response:

We corrected this sentence and several others.

Reviewer #3 (Remarks to the Author):

The manuscript is very much improved, and apart from some needed English editing in a few places, should be good to go. Congratulations to the authors on an excellent study!

Our response:

We really appreciate the reviewers' positive comments on and compliment to our work.

As responding to Reviewer #2, we edited the manuscript and Supplementary information.

A few comments:

1. Is it appropriate to refer to these iTALEs as pseudogenes? By definition, pseudogenes are functionless. In fact, this paper's major finding is that these genes are not pseudogenes.

Our response:

We refer these iTALEs as pseudogenes at the beginning of paper as they were annotated and reported as pseudogenes in prior studies. We then define and refer to them as iTALE after we demonstrate they are expressed and encode proteins. For the transition, we used this sentence: "The results indicate that the TALEs *Tal3a* and *Tal3b* are not pseudogenes as previously annotated but instead are expressed and function as TALE variants in PXO99^A for virulence by

interfering with the host resistance in IRBB1. Both effector variants and their relatives are referred to hereinafter as iTALEs (interfering TAL effectors).”

We think it is important to bring up the historical aspect of these genes and remind readers that annotated pseudogene(s) may not be functionless, an exciting point of our work.

2. Fig 1a – PthXo1 is said to have 24 repeats above the diagram; this should be changed to 23.5 repeats, which is shown with the diagram and the repeat sequence below the diagram.

Our response:

We changed that number for PthXo1 and the last repeat as 0.5 for all other TALEs.

3. Should NLR be defined in the introduction, not just in the abstract?

Our response:

We added the definition of NLR in Introduction.

4. It is unclear which cluster pthXo1 is from in Fig 1b. Can you indicate? Added in figure and indicated in legend.

Our response:

Actually, *pthXo1* corresponding to 2b (cluster 2) is already indicated in Fig. 1b and figure legend: “*pthXo1* corresponds to 2b.” To be clearer we changed the sentence in legend to “*pthXo1* corresponds to 2b of cluster 2”.

5. Comment: Fig 4b is impressive – much improved from the previous version.

No response.

6. Thanks for including Supplementary Fig. 16: some clarification: is this figure showing expression of the plasmid-borne Tal3b from PXO99A? and the native “B-type” iTALEs from each of the other strains?

Our response:

Expression of the plasmid-borne Tal3b from PXO99^A in any strain is not shown in Supp. Fig. 16. Only type B containing strains except PXO99^A (with both A and B type iTALE as a positive control) and T7174 (no any iTALE as a negative control) were used and included in Supp. Fig. 16. For clarification, we added this information in the legend.